# Phosphatidylcholine mediates the crosstalk between LET-607 and DAF-16 stress response pathways

**Bin He**, **Jie Xu**, **Shanshan Pang***, **Haiqing Tang***

School of Life Sciences, Chongqing University, Chongqing, China

* sspang@cqu.edu.cn (SP); hqtang@cqu.edu.cn (HT)

**Data Availability Statement:** The RNA sequencing data have been deposited in the GEO with an accession number of GSE155935. Other data are within the manuscript and its Supporting Information files.

## Abstract

Coordinated regulation of stress response pathways is crucial for cellular homeostasis. However, crosstalk between the different stress pathways and the physiological significance of this crosstalk remain poorly understood. In this study, using the model organism *C. elegans*, we discovered that suppression of the transcription factor LET-607/CREBH, a regulator of cellular defense and proteostatic responses, triggers adaptive induction of DAF-16-dependent stress responses. Suppression of LET-607 improves stress resistance and extends *C. elegans* lifespan in a DAF-16-dependent manner. We identified the sphingomyelin synthase SMS-5 to be a central mediator in the communication between LET-607 and DAF-16. SMS-5 reduces the contents of unsaturated phosphatidylcholine (PC), which activates DAF-16 through ITR-1-dependent calcium signaling and calcium-sensitive kinase PKC-2. Our data reveal the significance of crosstalk between different stress pathways in animal fitness and identify LET-607/CREBH and specific PC as regulators of DAF-16 and longevity.

## Author summary

In order to cope with stresses, cells have evolved complex and elegant adaptive mechanisms, which are also referred to as stress responses. Central to these responses are core transcription factors. It is widely hypothesized that interruption of one key stress response pathway could compromise overall cellular function and survival. In order to avoid such an issue, stress response pathways communicate with each other. A defect in one pathway may adaptively activate other pathways, thus restoring homeostasis and increasing fitness. However, how these pathways communicate is largely unexplored. In this study, we unraveled crosstalk between the LET-607 and DAF-16 pathways in *C. elegans*. Suppression of LET-607, a regulator of defense and proteostatic responses, was shown to adaptively activate DAF-16, which is a crucial regulator of general stress responses. This crosstalk was shown to be vital for animal fitness, as suppression of LET-607 extends lifespan in a DAF-16-dependent manner. Intriguingly, loss of LET-607 results in increased levels of the sphingomyelin synthase SMS-5, which metabolizes membrane lipid PC. Consequently, the reduction in PC causes activation of DAF-16 via membrane-located

**Funding:** This work was supported by National Natural Science Foundation of China (grant No. 32070754, 31701017 to H.T. and grant No. 31771337, 32071163 to S. P.), Natural Science Foundation of Chongqing, China (grant No. cstc2020jcyj-msxmX0714 to H.T.) and Chongqing Talents Plan for Young Talents (CQYC201905071 to S.P.). The funders had no role in study design, data collection and analysis, decision to publish, or preparation of the manuscript.

**Competing interests:** The authors have declared that no competing interests exist.

calcium channel ITR-1 and calcium-sensitive kinase PKC-2. This study identifies a novel crosstalk between stress response pathways, which is potentially significant in animal longevity.

## Introduction

In order to defend against environmental and cellular stresses, animals have evolved complex stress response pathways. Disturbance of these pathways leads to compromised cellular homeostasis and aging [1]. Stress responses typically activate in different parts of the cell, which are controlled by core transcription factors. These transcription factors each regulate a specific subset of genes that enable the cell to cope with compartment-specific stresses. Importantly, many stresses, such as heat and pathogens, can damage several cellular compartments. In order to cope with this, multiple stress response pathways must be activated simultaneously. Notably, Interruption of one stress response pathway could cause deleterious effects on other parts of the cell and as a result influence overall cellular homeostasis [2,3]. Therefore, disturbing one stress pathway may induce adaptive stress responses in other parts of the cell. This hypothesis has been corroborated by recent research investigating the communication between mitochondrial and cytosolic proteostatic response pathways. Specifically, mitochondrial stress has been shown to activate several homeostatic pathways in the cytosol to restore the cellular homeostasis [4–7]. Other than research into proteostatic pathways, little is known about whether and how stress pathways communicate to enable overall cellular homeostasis and animal fitness.

The endoplasmic reticulum (ER) is the main site for protein folding and lipid biosynthesis and a key organelle for cellular stress responses. One such response is the unfolded protein response (UPR), which restores proteostasis upon ER stress [8]. ER-resident transcription factors play a crucial role in ER stress responses. In mammals, CREBH is an ER-bound transcription factor and a member of the CREB3 family. Upon ER-stress, CREBH is activated and transits to the Golgi. It is subsequently cleaved by the proteases, after which the cleaved N-terminal fragment translocates to the nucleus, where it activates downstream target genes [9]. This process is known as regulated intramembrane proteolysis (RIP) [10]. CREBH activation promotes the expression of various inflammatory factors and iron-regulatory hormone hepcidin [9,11]. This links ER stress to cellular defense responses.

In this study, we used model organism *C. elegans* to study LET-607, the *C. elegans* ortholog of mammalian CREBH. This investigation revealed crosstalk between LET-607 and the cytosolic DAF-16 stress pathways. We showed that unlike mammalian CREBH, LET-607 is located in the nucleus and not on the ER. However, LET-607 regulates cellular defense genes as well as UPR genes like CREBH, thus indicating their functions may be conserved. In addition, suppression of LET-607 induces an adaptive activation of DAF-16, which is a master regulator of cytosolic stress responses in *C. elegans*. This activation results in improved animal health and longevity. We further uncovered the underlying mechanism of this crosstalk, involving the sphingomyelin synthase SMS-5, the membrane lipid PC, the ER-resident calcium channel inositol *1*,4,5-trisphosphate receptor (ITR-1) and the calcium-dependent kinase PKC-2. Together, these results delineate a cellular mechanism linking different stress response pathways and highlight its importance in animal longevity.

## Results

### LET-607 regulates multiple stress responses

Previously, we showed that constant activation of defense responses promotes *C. elegans* longevity [12]. In order to identify additional defense regulators, RNAi screening of the *C. elegans* transcription factor library was performed using the immune reporter *T24B8.5p*::GFP [13]. This identified *let-607* as a positive regulator of *T24B8.5p*::GFP (Fig 1A), which is consistent with the role of mammalian CREBH in innate immunity [9]. *let-607* RNAi arrested the development of *C. elegans*. In order to ensure animal development into adult, diluted *let-607* RNAi (1:5) was used. This reduced the *let-607* mRNA levels by about 75% (S1A Fig).

The mRNA expression of *let-607* is induced by ER stress [14], however its physiological functions in *C. elegans* remain largely unknown. Via the construction of a GFP-fused translational reporter LET-607::GFP, it was unexpectedly determined that LET-607 mainly localized in the nucleus, not on the ER membrane (Fig 1B). We further examined the colocalization of LET-607::GFP with DIS-3::mCHERRY. DIS-3 is a component of *C. elegans* nuclear RNA exosome complex that is enriched in the nucleoplasm but not in the nucleoli [15]. It was identified that LET-607::GFP overlapped well with DIS-3:: mCHERRY specifically in the nucleoplasm (Fig 1B). Protein sequence analysis showed that there was 40% homology between the transmembrane domains of LET-607 and CREBH (S1B Fig). These domains are critical for ER localization and differences in these domains may explain the nuclear localization of LET-607. This finding is also consistent with the localization of the *Drosophila* ortholog of CREBH, which lacks a transmembrane domain [16].

Next, the function of LET-607 was examined by collecting control RNAi and *let-607* RNAi-treated worms for RNA-sequencing analysis. This identified 860 genes regulated by LET-607, of which 316 were upregulated and 544 were downregulated (S1 Table). Genes which were decreased by *let-607* RNAi were classified as upregulated by LET-607. Functional analysis revealed these genes were enriched in immune and detoxification responses (Fig 1C). These upregulated genes included C-type lectins and innate immunity genes, which are known to play a key role in pathogen defenses. These findings are again consistent with the function of mammalian CREBH. In addition, cytochrome P450, which mediates detoxification response, and peptidases, which are critical for cellular proteostasis, were also upregulated. Intriguingly, as shown in S1 Table, the basal expression of the classic ER UPR genes, including *hsp-3* and *hsp-4*, was not significantly affected by *let-607* RNAi. However, ER stress-induced HSP-4 activation was suppressed by *let-607* RNAi (S1C Fig). This is in line with a previous finding suggesting that LET-607 binds to the promoter of *hsp-4* [17]. Together, these results indicate that LET-607 positively regulates genes involved in multiple stress responses, specifically the defense and proteostatic responses. This suggests LET-607 functions are in some part conserved with mammalian CREBH.

Next, the association between the identified genes and altered stress resistance was assessed. The stresses tested included *Pseudomonas aeruginosa* strain PA14-induced pathogen stress, tert-butyl hydroperoxide (TBHP)-induced xenobiotic stress, 35˚C heat stress and dithiothreitol (DTT)-induced ER stress. Unexpectedly, although LET-607 positively regulated stress response genes, *let-607* RNAi increased *C. elegans* survival in response to all of the stresses tested (Fig 1D–1G), suggesting that *let-607* knockdown increases the fitness of *C. elegans* upon stress. In addition, TBHP and heat stress were used to test tissue specificity. It was identified that intestine-specific *let-607* RNAi improved animal stress resistance, whereas other tissue-specific RNAis (specifically germline, muscle and hypodermis) had no such effects (S1D–S1K Fig). As stress resistance is known to contribute to longevity [18], animal lifespan was measured. It was observed that *let-607* RNAi indeed significantly extended *C. elegans* lifespan (Fig

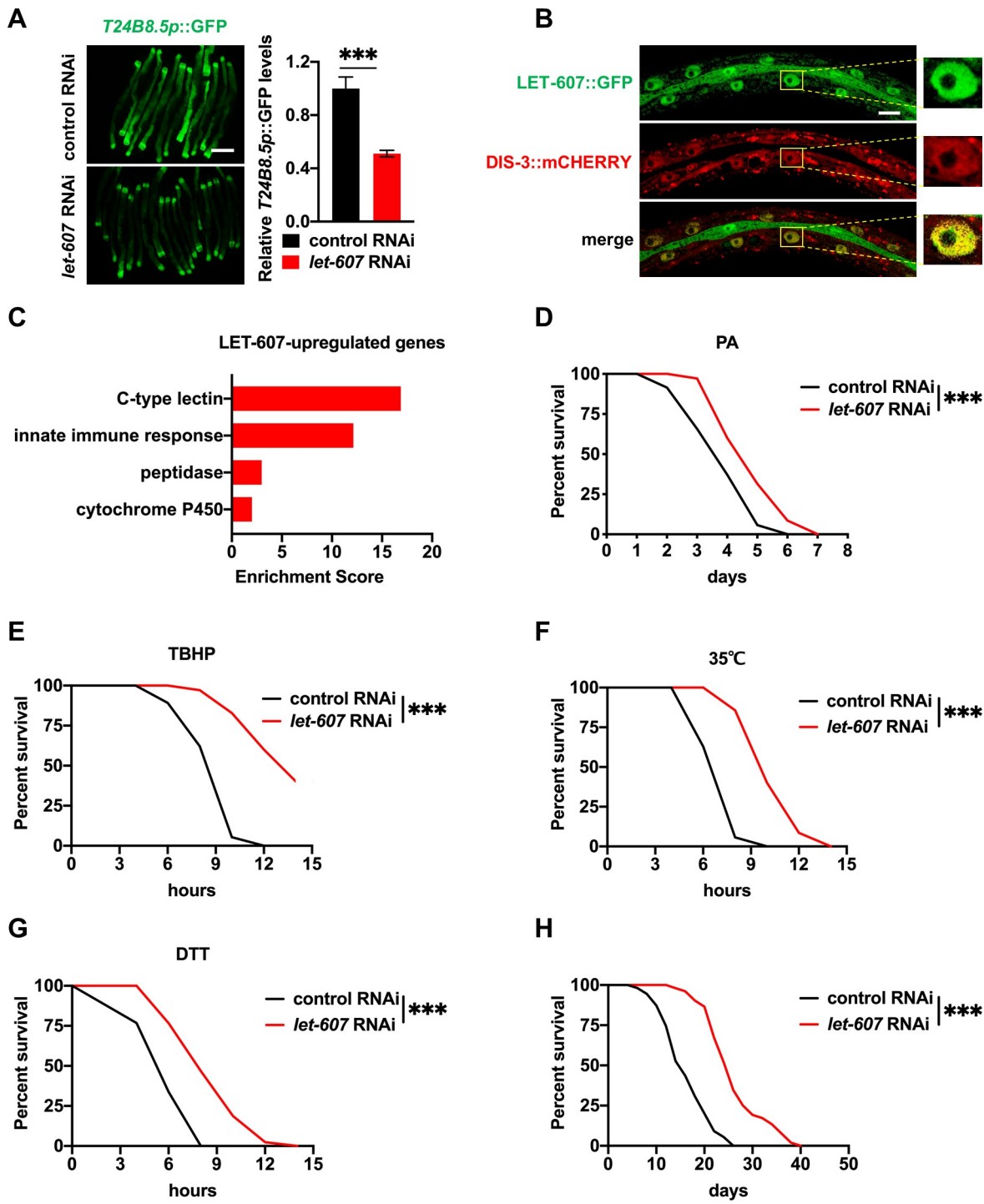

**Fig 1. LET-607 regulates stress responses.** (A) Effects of *let-607* RNAi on *T24B8.5p*::GFP expression at 25°C. Left panel, representative images. Right panel: quantification data. Scale bar = 100 μm. (B) Nuclear expression of LET-607::GFP. LET-607::GFP overlap with DIS-3:: mCHERRY in the nucleoplasm. Scale bar = 10 μm. (C) Functional classification analysis of genes upregulated by LET-607. (D-G) Effects of *let-607* RNAi on pathogen resistance (D), TBHP resistance (E), 35°C heat stress resistance (F) and DTT resistance (G). (H) Effects of *let-607* RNAi on *C. elegans* lifespan. Data were presented as mean ± SEM. *** p < 0.001 versus empty vector controls.

1H). Since *let-607* RNAi compromised animal development, the effects of post-developmental RNAi were also tested. The results showed that post-developmental treatment of *let-607* RNAi could still enhance stress resistance and extend lifespan (S1L–S1N Fig), suggesting that the effects of LET-607 in development and longevity may represent antagonistic pleiotropy. These data suggest that LET-607 negatively regulates animal stress resistance and longevity. Since stress resistance is determined by the outcome of multiple stress response pathways, we hypothesized that *let-607* RNAi might adaptively activate other pathways that together account for the enhanced stress resistance and longevity.

## LET-607 suppression activates stress response factor DAF-16

Previous studies have reported that pathogen resistance in *C. elegans* is governed by several cytosolic transcription factors including DAF-16 [19], HSF-1 [20] and SKN-1 [21,22]. Heat shock response is also controlled by HSF-1. Previously, *let-607* RNAi was shown to promote the expression of HSF-1 targets [23]. We speculated that *let-607* RNAi might activate the aforementioned factors and therefore enhance stress resistance. The results showed that the expression of the DAF-16 reporter MTL-1::GFP (Fig 2A) and the HSF-1 reporter *hsp-16.2p*::GFP [24] (S2A Fig) was increased by *let-607* RNAi. Contrastingly, the SKN-1 reporter *gst-4p*::GFP [25] was unaffected (S2B Fig).

We focused on DAF-16 in the following study. The induction of DAF-16 target genes was confirmed via use of another reporter, *sod-3p*::GFP (S2C Fig), and via *sod-3* mRNA expression (Fig 2B). The mRNA levels of *daf-16* were unchanged by *let-607* RNAi (Fig 2B). Upon activation, DAF-16 translocates to the nucleus. Indeed, enhanced nuclear occupancy of DAF-16 in *let-607* RNAi-treated worms was observed (Fig 2C). These findings suggest that suppression of LET-607 activates the cytosolic stress response factor DAF-16. As LET-607/CREBH could respond to ER stress [9,14] and regulate ER UPR genes [9] (S1C Fig), we further tested whether suppression of the classic ER stress response pathways could also activate DAF-16. This was investigated using RNAi of the ER UPR genes (*ire-1*, *pek-1* and *atf-6*). We found that despite suppressing downstream UPR genes as expected (S2D Fig), these RNAi did not affect MTL-1::GFP expression (Fig 2D). This suggests that suppression of LET-607 activates DAF-16 in a manner independent of ER UPR. Mammalian CREBH could be regulated by the core clock oscillator BMAL1 and glycogen synthase kinase 3β (GSK3β) [26]. However, knockdowns of the orthologs of these genes (*aha-1* and *gsk-3*) (S2E Fig) had no significant effects on MTL-1::GFP expression (S2F and S2G Fig), suggesting these genes may not be involved in DAF-16-induction. This is likely because of the nuclear localization of LET-607, whereas BMAL1 and GSK3β regulate ER-to-Golgi transport and subsequent proteolytic cleavage of CREBH [26]. In addition, PPARα has been reported to interact with CREBH to regulate target genes [27]. In this study, mutation of *nhr-49*, the functional ortholog of *pparα*, only marginally induced MTL-1::GFP expression in both control RNAi and *let-607* RNAi-treated animals (S2H Fig), suggesting NHR-49 is not involved in LET-607-mediated regulation of DAF-16.

Next, we predicted that *let-607* RNAi may also activate DAF-16-dependent stress response genes. Previously, a study classified DAF-16 responsive genes as 1,663 DAF-16-positive targets (class I) and 1,733 DAF-16-negative targets (class II) [28]. The identified *let-607* RNAi-induced genes were compared with the DAF-16 class I targets, and it was observed that 74 genes upregulated by *let-607* RNAi (13.6%) belonged to the DAF-16 class I targets (S1 Table). This percentage is significantly greater than what would be expected by chance (Fig 2E). This further supports the hypothesis that *let-607* RNAi regulates DAF-16. Weicksel et al. used ChIP-seq analysis to identify genes possessing LET-607 binding sites within promoter regions. Here, we compared the above 74 overlapped genes with their findings and found only one

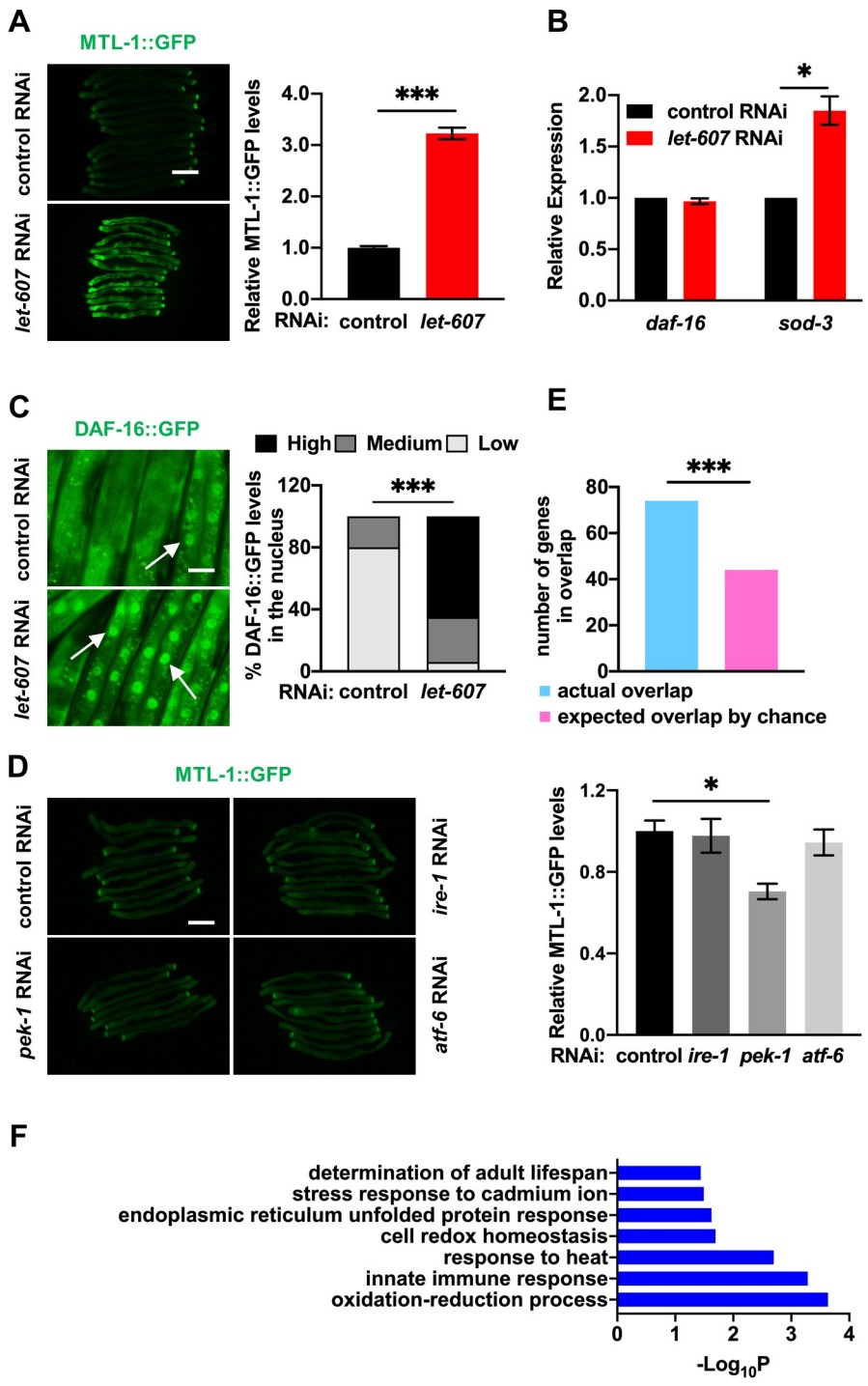

**Fig 2. Suppression of LET-607 adaptively activates DAF-16.** (A) Effects of *let-607* RNAi on MTL-1::GFP expression. Left panel, representative images. Right panel: quantification data. Scale bar = 100 μm. (B) Effects of *let-607* RNAi on the mRNA expression of *daf-16* and *sod-3* as measured by qPCR. n = 3 per group. (C) Effects of *let-607* RNAi on DAF-16::GFP nuclear occupancy. Left panel, representative images. White arrow indicates the nuclear GFP signal. Right panel: semi-quantification data. Scale bar = 25 μm. Number of animals (n): control RNAi (85) and *let-607* RNAi (96). (D) RNAi of the ER UPR genes had no effects on MTL-1::GFP expression. Left panel, representative images. Right panel: quantification data. Scale bar = 100 μm. (E) Number of genes that were induced by *let-607* RNAi and also DAF-16 class I target genes. *** p < 0.001 versus actual overlap. (F) GO term analysis of *let-607* RNAi-induced DAF-16 class I genes. Data were presented as mean ± SEM. * p < 0.05, *** p < 0.001.

gene contained LET-607 binding sites, suggesting LET-607 does not regulate these genes directly and likely functions via DAF-16. In addition, Gene Ontology (GO) analysis of the biological process revealed that these overlapped genes were enriched in several categories, including multiple stress responses, oxidation-reduction and lifespan determination (Fig 2F), consistent with the phenotype of the *let-607* knockdown. The adaptive induction of the DAF-16 downstream stress response genes may be the underlying mechanism conferring enhanced stress resistance in *let-607* RNAi-treated animals.

## LET-607 regulates stress resistance and longevity via DAF-16

Next, the role of DAF-16 in enhanced stress resistance was investigated in the *let-607* RNAi-treated animals. Mutation of *daf-16* abrogated or at least partially suppressed animal resistance to pathogenic and xenobiotic stresses (Fig 3A and 3B). However, in the case of *hsf-1* mutant animals, *let-607* RNAi enhanced the survival in the presence of these two stresses (S3A and S3B Fig). This suggests that resistance to pathogenic and xenobiotic stresses requires DAF-16. Contrastingly, enhanced heat stress resistance did not require DAF-16 (S3C Fig) but was in fact partially dependent on HSF-1 (S3D Fig). In summary, these data imply that in *let-607* knockdown animals, DAF-16 is mainly responsible for the defense response while HSF-1 contributes to the proteostatic response. This supports the notion that adaptive activation of other stress response pathways contributes to the enhanced stress resistance.

As DAF-16 is a crucial regulator of longevity, we further examined the role of DAF-16 induction in this process. As expected, the extended lifespan of *let-607* RNAi-treated worms was largely abolished by a loss-of-function mutation of *daf-16* (Fig 3C). Previous studies have shown that DAF-16 is essential for the longevity of some well-known long-lived models, including the insulin/IGF receptor *daf-2* mutants [29–31] and germline-less *glp-1* mutants [32,33]. In this study, we identified that *let-607* knockdown further increased the lifespan of *daf-2* mutants (S3E Fig), but failed to influence the lifespan of *glp-1* mutants (Fig 3D), suggesting that the mechanisms conferring lifespan extension in *let-607* knockdown and germline-deficient models are shared. Next, the impact of *let-607* knockdown on reproductivity was investigated. The numbers of progeny were found to be significantly reduced (S3F Fig), consistent with a previous report [17]. However, genes critical for longevity in germline-deficient mutants, including *daf-9*, *daf-12*, *tcer-1* and *kri-1* [33–35], were dispensable for lifespan extension in *let-607* RNAi animals (Fig 3E–3H), suggesting that *let-607* knockdown does not promote longevity via activation of the germline pathway.

## The sphingomyelin synthase SMS-5 mediates DAF-16 activation

Using RNAi screening of the LET-607-regulated genes identified in RNA-seq analysis, we further explored the specific mechanism of DAF-16 activation. This identified *C. elegans* sphingomyelin synthase SMS-5 as a new regulator of DAF-16. *sms-5* RNAi (S4A Fig) inhibited the activation of MTL-1::GFP upon *let-607* knockdown (Fig 4A). *sms-5* mRNA levels were upregulated by *let-607* RNAi (Fig 4B). Nuclear localization of DAF-16::GFP, xenobiotic stress resistance and lifespan extension induced by *let-607* RNAi were either inhibited or partially weakened via *sms-5* RNAi (Figs 4C, S4B and S4C), suggesting SMS-5 is required for DAF-16 induction.

Sphingomyelin synthase catalyzes the production of diacylglycerol (DAG) and sphingomyelin at the expense of ceramide and PC (S4D Fig). We hypothesized that one of these membrane lipids might cause activation of DAF-16. In order to test this hypothesis, we first examined ceramides, which are key lipid signaling molecules in multiple cellular processes.

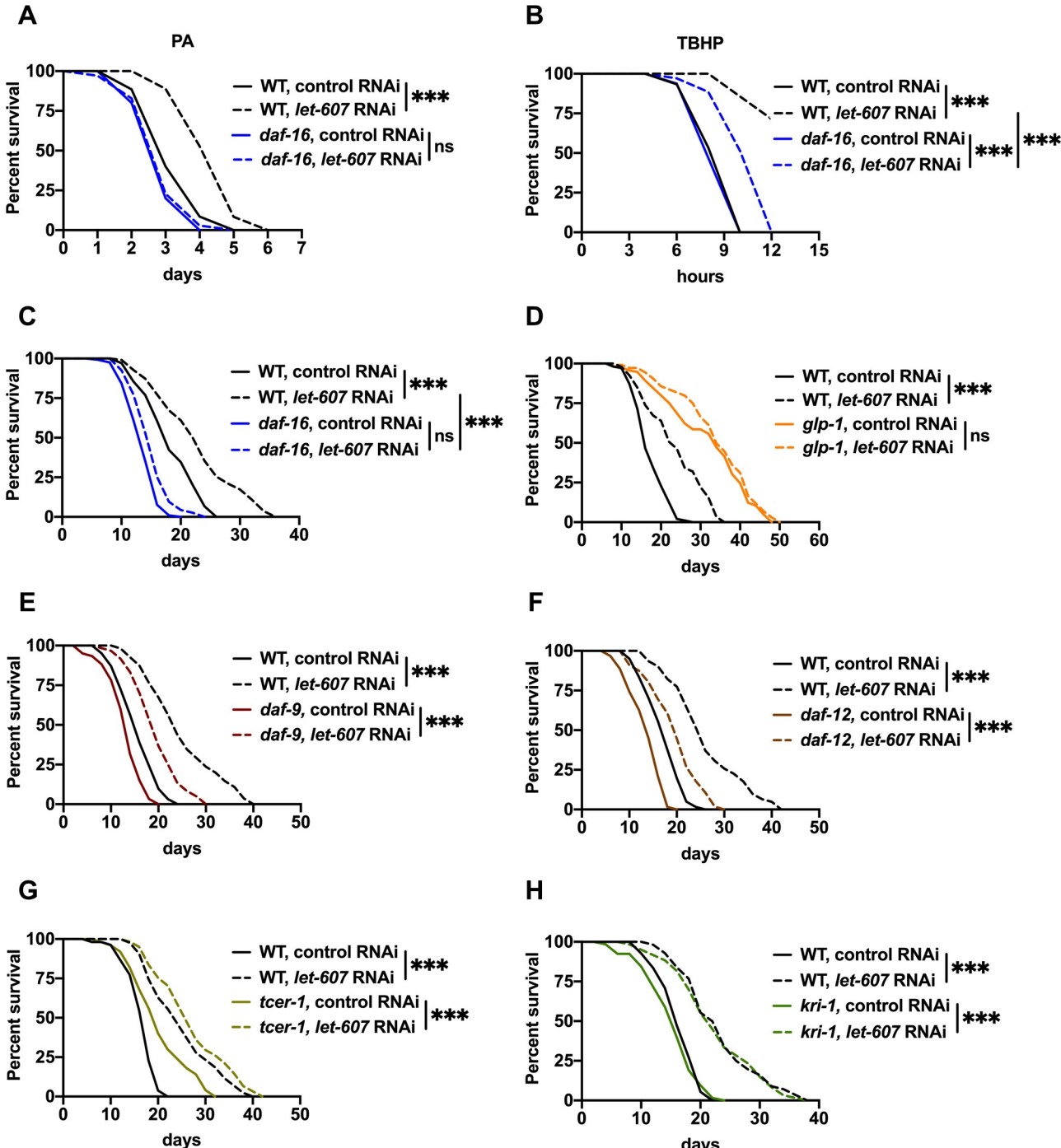

**Fig 3. Suppression of LET-607 increases stress resistance and extends lifespan via DAF-16.** (A-C) Effects of *let-607* RNAi on pathogen resistance (A), TBHP resistance (B), and lifespan (C) in wild-type (WT) and *daf-16* mutant worms. (D) Effects of *let-607* RNAi on the lifespan of *glp-1* mutants. (E-H) Effects of *let-607* RNAi on the lifespan of *daf-9* (E), *daf-12* (F), *tcer-1* (G) and *kri-1* (H) mutants. *** p < 0.001 versus empty vector controls.

Supplementation of ceramides with various acyl chains did not affect the expression of MTL-1::GFP induced by *let-607* RNAi (S4E Fig), suggesting that ceramide is not likely to be involved in DAF-16 induction.

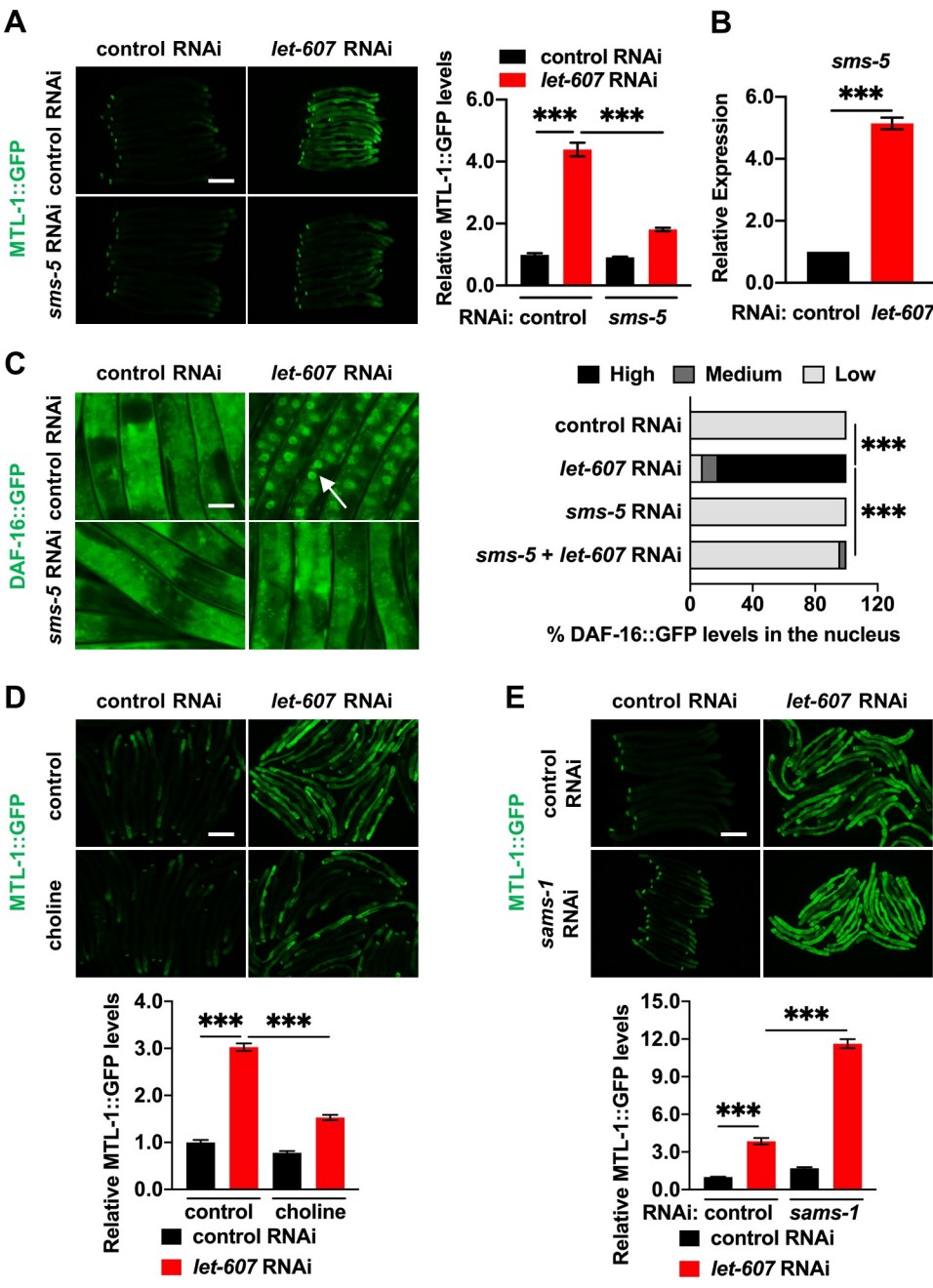

**Fig 4. The sphingomyelin synthase SMS-5 mediates DAF-16 activation upon *let-607* knockdown.** (A) Effects of *sms-5* RNAi on *let-607* RNAi-induced MTL-1::GFP expression. Left panel, representative images. Right panel: quantification data. Scale bar = 100 μm. (B) Effects of *let-607* RNAi on the *sms-5* mRNA expression. n = 3 per group. (C) Effects of *sms-5* RNAi on *let-607* RNAi-induced DAF-16::GFP nuclear occupancy. Left panel, representative images. White arrow indicates nuclear GFP signal. Right panel: semi-quantification data. Number of animals (n): control RNAi (97), *let-607* RNAi (87), *sms-5* RNAi (80) and *sms-5* + *let-607* RNAi (48). Scale bar = 25 μm. (D) Influence of choline on MTL-1::GFP expression in *let-607* RNAi-treated animals. (E) MTL-1::GFP expression in WT and *let-607* knockdown animals after *sams-1* RNAi. Upper panel, representative images. Lower panel: quantification data. Scale bar = 100 μm. Data were presented as mean ± SEM. *** p < 0.001.

We next tested whether the consumption of PC by SMS-5 might activate DAF-16. Dietary supplementation of choline, a metabolic precursor for PC, suppressed the expression of MTL-1::GFP upon *let-607* knockdown (Fig 4D). In *C. elegans*, PC is synthesized in a variety of ways including from choline (via the Kennedy pathway) or via the action of SAMS-1 and PMT-1/PMT-2-mediated sequential methylation from phosphoethanolamine (S5A Fig). We next asked whether PC reduction was sufficient to activate DAF-16 by silencing the genes required for PC synthesis. However, RNAi knockdown of PC biosynthetic genes, including *sams-1*, *pmt-2*, *ckb-1*, *pcyt-1* and *cept-1* (S5B Fig), did not cause any significant changes in MTL-1::GFP expression (Figs 4E and S5C). Subsequently, we investigated the synergetic effects of *let-607* and PC biosynthetic genes. The combined knockdown of *let-607* and PC synthetic genes via double RNAi (S5B Fig) synergistically activated MTL-1::GFP (Figs 4E and S5C), suggesting that PC reduction participates in the activation of DAF-16. It was also noted that the expression of PC biosynthetic genes was reduced after *let-607* RNAi (S5B Fig), further suggesting that *let-607* knockdown may suppress PC production. These data are consistent with a model that reduction of PC is essential for DAF-16 activation in response to *let-607* knockdown.

## LET-607 regulates PC metabolism

As it was established that LET-607 regulates DAF-16 via PC, we investigated the function of LET-607 in lipid metabolism. Using GC-MS/MS, a decrease in variety of fatty acid species was identified after *let-607* RNAi (Fig 5A), which resulted in a decline in the overall concentrations of both saturated and unsaturated fatty acids (S6A Fig). Fittingly, triglyceride contents were also decreased in *let-607* RNAi worms (S6B Fig). We next examined the membrane lipid PC and found that *let-607* RNAi significantly decreased PC contents (Fig 5B), which was mostly attributed to the reduction of unsaturated PC (Fig 5C). Other major membrane phospholipids were also examined. Phosphatidylethanolamine (PE) and phosphatidylserine (PS) were shown to not be significantly affected, whereas the levels of phosphatidylinositol (PI), especially unsaturated PI, were greatly reduced (Fig 5B and 5C). The PC/PE ratio, which influences membrane fluidity and curvature, was found to not change significantly in *let-607* RNAi-treated animals (S6C Fig).

Further analysis of the fatty acid composition of PC revealed that the levels of numerous unsaturated fatty acids in PC were indeed greatly reduced (Fig 5D). As a control, most unsaturated fatty acids in PE fraction showed no significant changes (Fig 5D), again supporting the importance of LET-607 in PC regulation. As unsaturated PC was significantly reduced by *let-607* knockdown, worms treated with *let-607* RNAi were directly supplemented with unsaturated PC (18:1n9). This supplementation suppressed the induction of MTL-1::GFP (Fig 5E and 5F), which phenocopied the *sms-5* RNAi and suggest unsaturated PC could indeed regulate DAF-16.

## PC mediates DAF-16 activation in germline-deficient mutants

Having established SMS-5 and PC reduction as mediators of DAF-16 activation, our next question was whether they are generally required for DAF-16 activation in other models, including *daf-2* mutants, *glp-1* mutants, heat stress and oxidative stress. Supplementation of unsaturated PC (18:1n9) partially suppressed MTL-1::GFP induction specifically in *glp-1* mutants (Fig 6A). This was not the case for *daf-2* mutants (Fig 6B), or worms subjected to heat stress (Fig 6C) or oxidative stress (Fig 6D). Similar results were also observed for the nuclear occupancy of DAF-16 in *glp-1* and *daf-2* mutants (S7A and S7B Fig), suggesting that PC specifically mediates DAF-16 activation in germline-deficient mutants. This is consistent with the lifespan data (Fig 3D). As for *sms-5*, it was determined that its RNAi had no significant impact

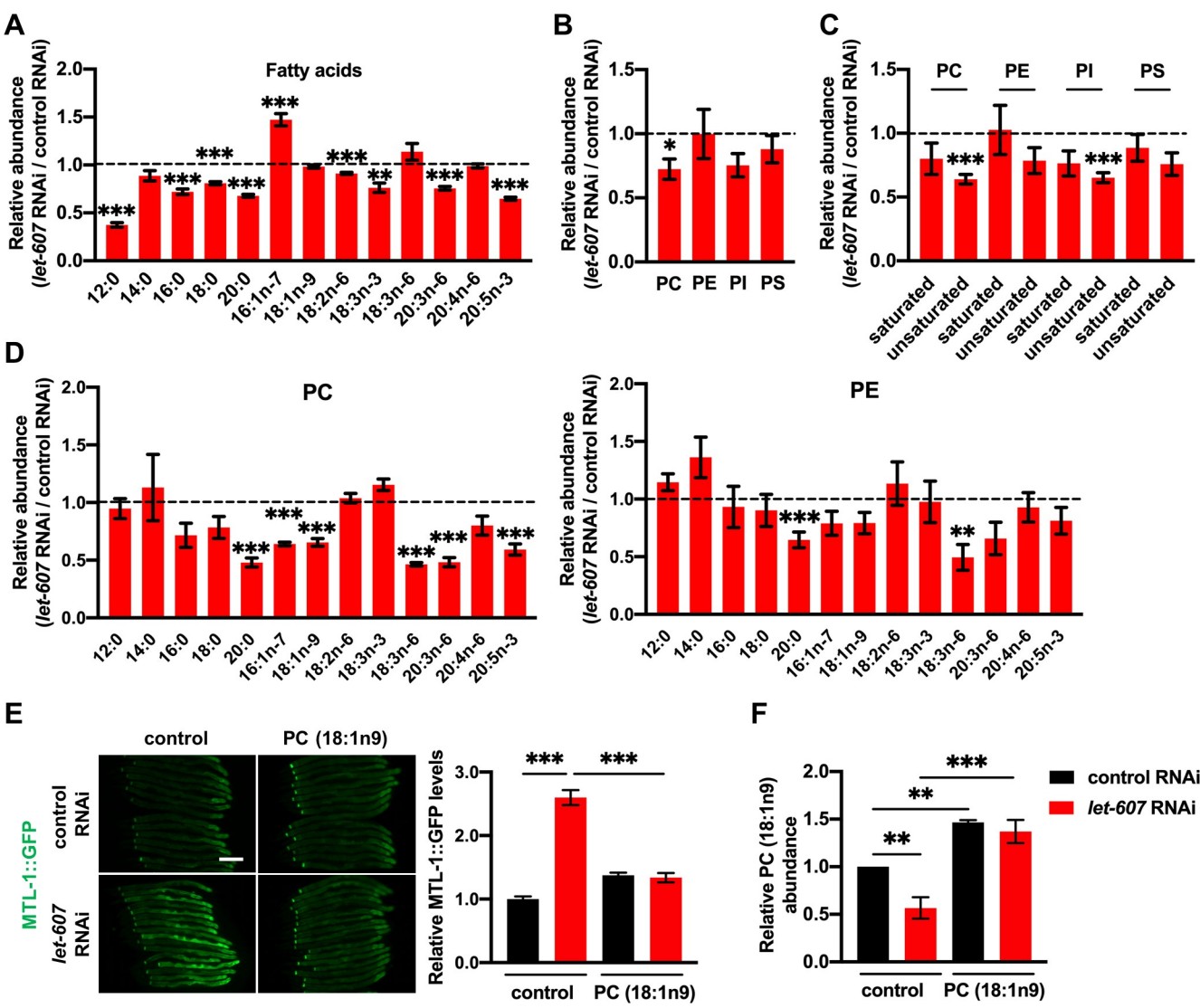

**Fig 5. LET-607 regulates PC metabolism.** (A) Effects of *let-607* RNAi on fatty acid contents. (B) Effects of *let-607* RNAi on the abundance of membrane lipids. (C) Effects of *let-607* RNAi on the abundance of saturated and unsaturated fatty acids in the PC, PE, PI and PS fractions. (D) Fatty acid contents in the PC and PE fractions after *let-607* RNAi. (E) MTL-1::GFP expression in *let-607* knockdown animals after PC supplementation. Left panel, representative images. Right panel: quantification data. Scale bar = 100 μm. (F) PC (18:1n9) contents after PC (18:1n9) supplementation. For (A) and (F), n = 3 per group; for (B-D), n = 5 per group. Data were presented as mean ± SEM. * $p < 0.05$, ** $p < 0.01$, *** $< 0.001$.

on MTL-1::GFP induction in any of the long-lived models (Fig 6E and 6F) or under stressed conditions (Fig 6G and 6H). This suggests that in *glp-1* mutants, SMS-5 does not control PC metabolism and other mechanisms are at play.

## LET-607 regulates DAF-16 via calcium signaling

PC is one of the most abundant membrane lipids and has significant influence on the membrane proteins. PC composition influences membrane properties that in turn alter the functions of the membrane proteins, such as the ER membrane-located proteins [36,37]. We therefore speculated that some membrane proteins mediate the activation of DAF-16. The ER-located calcium channel and calcium homeostasis are regulated by alterations in PC contents

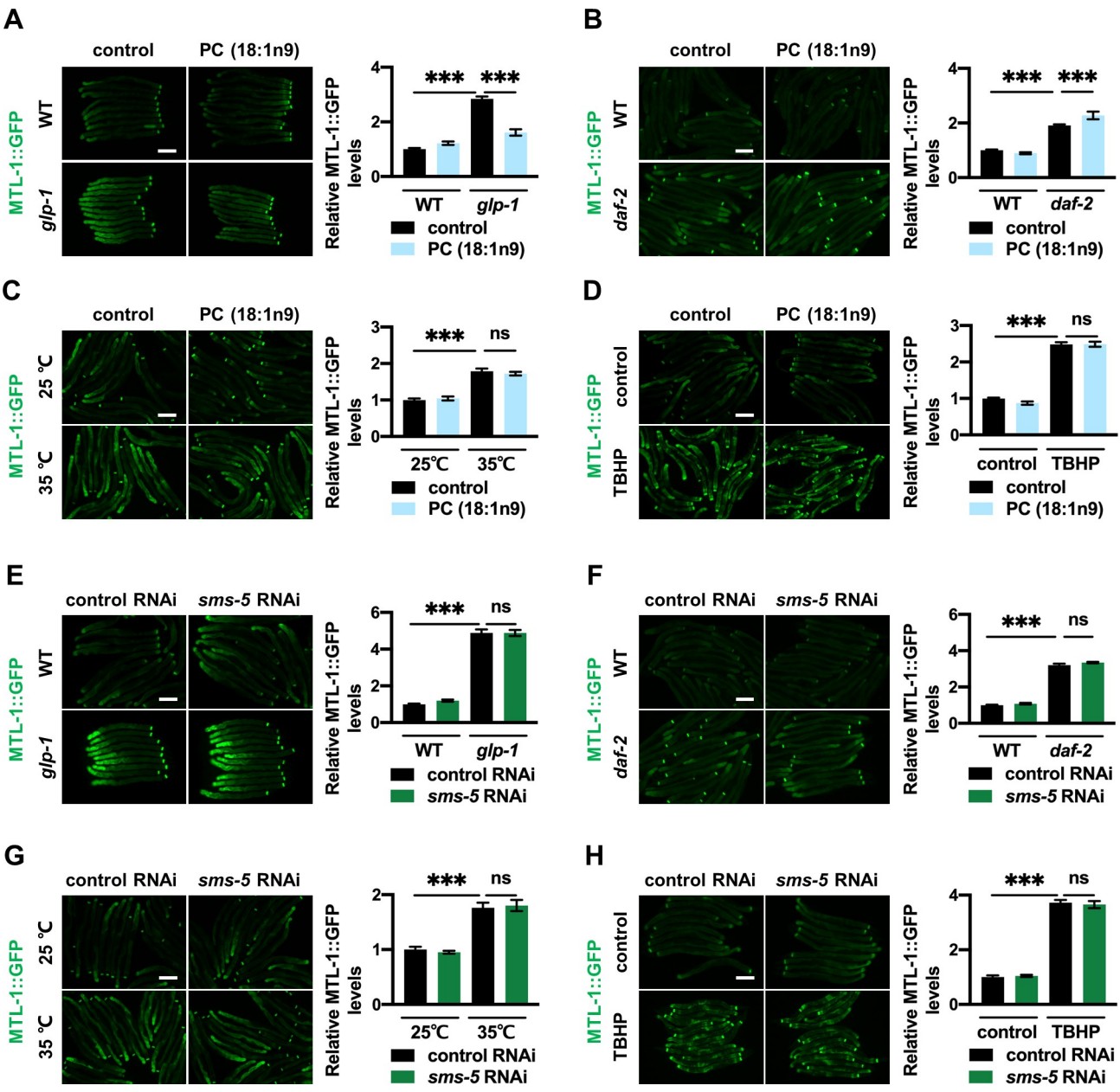

**Fig 6. PC mediates DAF-16 activation in *glp-1* mutants.** (A-D) Effects of PC (18:1n9) supplement on MTL-1::GFP induction in *glp-1* mutants (A), *daf-2* mutants (B), 35°C heat stressed worms (C) and TBHP-treated worms (D). (E-H) Effects of *sms-5* RNAi on MTL-1::GFP induction in *glp-1* mutants (E), *daf-2* mutants (F), 35°C heat stressed worms (G) and TBHP-treated worms (H). Left panel, representative images. Right panel: quantification data. Scale bar = 100 μm. Data were presented as mean ± SEM. *** p < 0.001.

[37]. What's more, calcium has been reported to regulate DAF-16 activity [38]. Thus, we asked whether DAF-16 induction requires the ER-resident calcium channels. The role of sarco/endoplasmic reticulum $Ca^{2+}$-ATPase (SERCA) is to pump calcium into the ER from the cytosol. Conversely, inositol 1,4,5-trisphosphate (IP3) receptor releases calcium from the ER to the cytosol. In *C. elegans*, SERCA and the IP3 receptor are encoded by *sca-1* and *itr-1*, respectively. We found that *itr-1* RNAi (S8A Fig) largely abrogated MTL-1::GFP activation upon *let-607* knockdown (Fig 7A). Furthermore, a loss-of-function mutation of *itr-1*, which results in

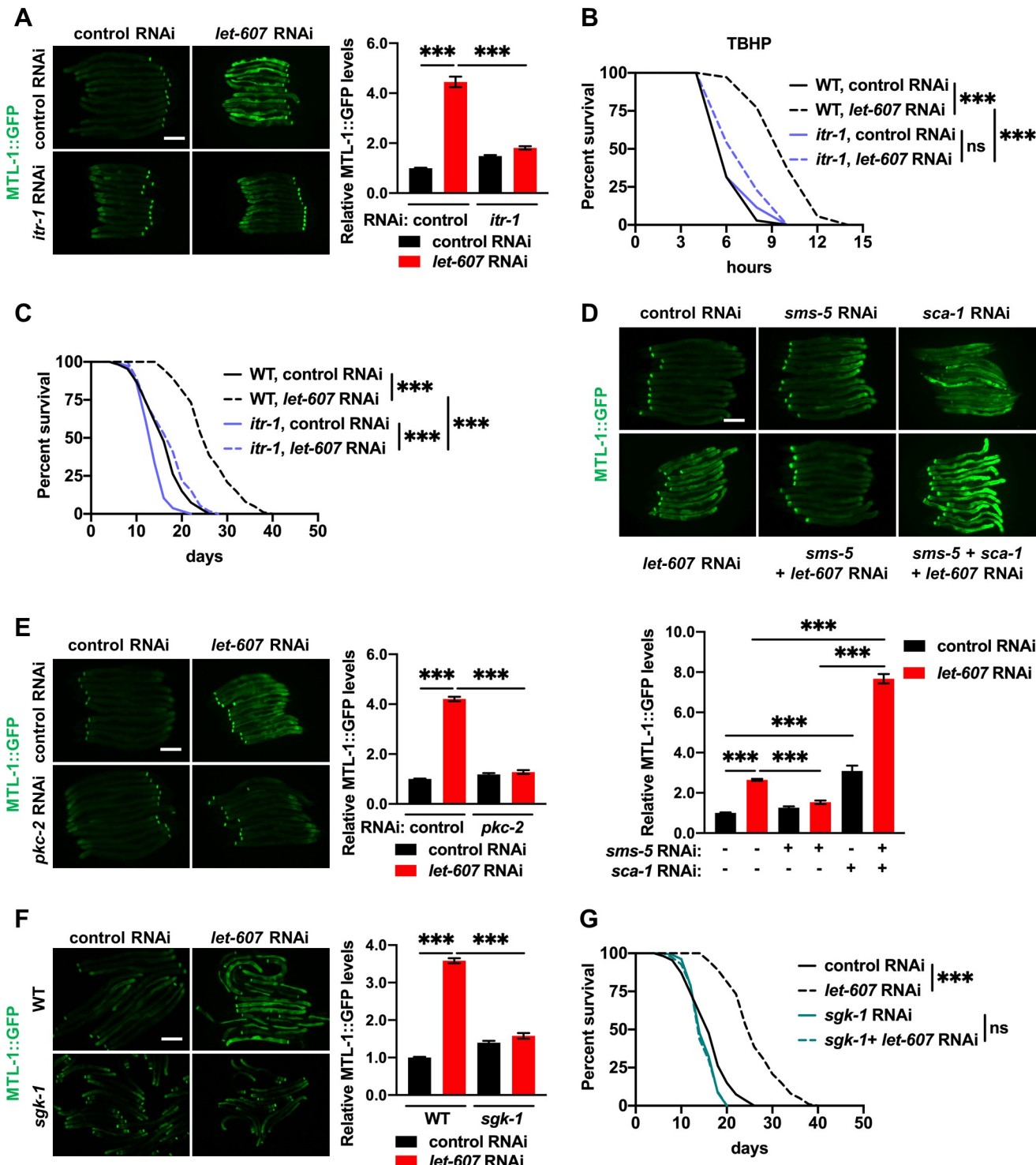

**Fig 7. Crosstalk between LET-607 and DAF-16 depends on calcium signaling and calcium-sensitive kinase.** (A) Effects of *itr-1* RNAi on *let-607* knockdown-induced MTL-1::GFP expression. Left panel, representative images. Right panel: quantification data. Scale bar = 100 μm. (B-C) Effects of *itr-1* mutation on *let-607* RNAi-induced TBHP resistance (B) and longevity (C) in *C. elegans*. (D) Effects of *sca -1* RNAi on the expression of MTL-1::GFP which is suppressed by *sms-5* RNAi. Upper panel, representative images. Lower panel: quantification data. Scale bar = 100 μm. (E-F) Effects of *pkc-2* RNAi (E) and *sgk-1* mutation (F) on *let-607* knockdown-induced MTL-1::GFP expression. Left panel, representative images. Right panel: quantification data. Scale bar = 100 μm. (G) Influence of *sgk-1* RNAi on the lifespan of WT and *let-607* knockdown worms. Data were presented as mean ± SEM. *** p < 0.001.

defects in ER-derived calcium signaling, was found to significantly reduce the enhanced stress resistance (Fig 7B) and extended lifespan (Fig 7C) of *let-607* RNAi-treated animals. We also examined the effect of *egl-8* RNAi, which compromises ITR-1-dependent calcium signaling through the reduction of the ITR-1 ligand IP₃. The results showed that *egl-8* knockdown (S8B Fig) completely abrogated the effects of *let-607* RNAi on MTL-1::GFP expression (S8C Fig) and xenobiotic stress resistance (S8D Fig), which phenocopied *itr-1* mutation. These data suggest that LET-607-mediated DAF-16 activation requires the ITR-1 calcium channel.

As for *sca-1* RNAi (S8E Fig), we found that it increased MTL-1::GFP intensity, and importantly, *sca-1* RNAi restored MTL-1::GFP expression with *sms-5* RNAi (Fig 7D), indicating that the calcium signal acts downstream of SMS-5. Similarly, the addition of ionomycin, which is a calcium ionophore that induces intracellular calcium flux, restored the expression of MTL-1::GFP in response to *sms-5* RNAi (S8F Fig).

Calcium has been reported to regulate DAF-16 activity via calcium-sensitive kinase PKC-2 and the downstream kinase SGK-1 [38]. In line with this, both *pkc-2* RNAi (S8G Fig) and *sgk-1* mutation abolished the enhanced MTL-1::GFP expression in *let-607* RNAi-treated worms (Fig 7E and 7F). Accordingly, *sgk-1* inhibition also suppressed the stress resistance (S8H Fig) and longevity (Fig 7G) observed upon *let-607* knockdown. We also tested mutation of *rict-1*, a TORC2 component and well-recognized upstream regulator of *sgk-1* in *C. elegans* [39,40]. Unlike *sgk-1*, it did not suppress MTL-1::GFP induction in *let-607* RNAi-treated worms (S8I Fig), suggesting TORC2 is not involved. DAF-16 can also be phosphorylated and suppressed by AKT [32,41]. Consistently, *akt-1;akt-2(RNAi)* enhanced MTL-1::GFP expression. They did however have similar effects in both wild type and *let-607* RNAi animals (S8J Fig), suggesting that AKT is not relevant to LET-607-mediated DAF-16 activation. Together, these data suggest that *let-607* RNAi-mediated DAF-16 activation requires PKC-2 and SGK-1 kinases.

## Discussion

This study unraveled crosstalk between LET-607 and DAF-16 stress response pathways. This crosstalk is important for animal fitness, as evidenced by the enhanced stress resistance and extended lifespan. Furthermore, this crosstalk was shown to be regulated via SMS-5-mediated reduction of PC, which modulates the DAF-16 pathway through ITR-1-dependent calcium signaling and downstream kinases PKC-2 and SGK-1 (Fig 8).

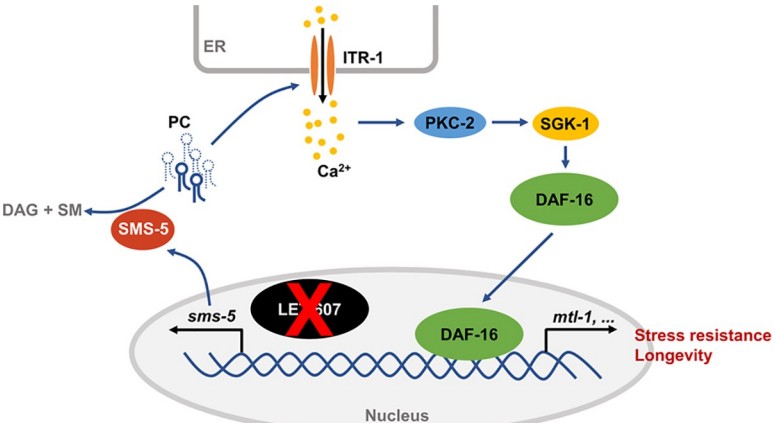

**Fig 8. Communication model between the LET-607 and DAF-16 pathways.** Suppression of LET-607 induces the expression of *sms-5*, which reduces unsaturated PC contents. This may in turn affect the ITR-1 calcium channel on the ER membrane. Calcium released by ITR-1 activates DAF-16 via calcium-sensitive kinase PKC-2 and downstream kinase SGK-1.

DAF-16 induction is specific for LET-607, but not for classic ER UPR factors IRE-1, PEK-1 and ATF-6. The function of mammalian CREBH is to regulate the inflammatory genes and hormone hepcidin [9,11]. Our study revealed that *C. elegans* LET-607 induces several important defense genes such as C-type lectins and cytochrome P450. These findings together suggest a critical role for LET-607/CREBH in promoting cellular defense responses against pathogens and xenobiotic molecules. Importantly, DAF-16 is a major transcription factor in *C. elegans*, which controls various responses including detoxification and pathogen responses [19,28,42]. This is in accordance with the adaptive activation of DAF-16 when LET-607 is compromised.

The ER plays a central role in coordinating cellular stress responses. ER stress can initiate multiple defensive or homeostatic processes aimed at restoring function of not only the ER but also the whole cell. Key mediators linking the ER and other parts of the cell are the ER stress-responsive proteins, which include the classic ER UPR factors (IRE1, PERK and ATF6) and CREBH/LET-607 [3,43,44]. It is plausible that when these ER stress response factors are compromised, adaptive responses are triggered across the cell to restore homeostasis. Although LET-607 is localized in the nucleus, it is regulated by ER stress [14]. And the function of LET-607/CREBH appears to be much conserved, with a key role being to control the immune-related gene expression. The communication between LET-607 and DAF-16 may be an adaptive response specifically compensating for compromised LET-607-mediated ER stress response. Further studies are required to identify the components of the LET-607-dependent ER stress response pathway.

Previous studies have shown that CREBH plays a crucial role in glucose and lipid metabolism [44,45]. This study identified previously unknown functions of LET-607, namely in the metabolism of membrane lipids. We hypothesize that LET-607 may respond to membrane lipid disturbance and may play a role in the maintenance of membrane homeostasis. This idea warrants further study. This study also found that LET-607 regulates SMS-5, which works to decrease the ceramide contents and increase the levels of DAG and sphingomyelin. These membrane lipids are critical mediators of multiple cellular processes, implying that LET-607 may have a broader role in metabolic and cellular homeostasis.

DAF-16 induction and longevity are regulated by SMS-5 and the membrane lipid PC. The central roles of membrane lipids in animal aging are only beginning to emerge. Studies in humans as well as model organisms have revealed that aging is associated with significant changes in multiple membrane lipid classes [46]. Specifically, the contents of many PC species have been found to increase with the animal aging progress [47,48], which is consistent with our observation that the reduction of PC is beneficial for animal health. It is not yet fully understood how PC regulates the aging process. This study showed that DAF-16 activation is mediated by the ER-resident calcium channel ITR-1. Specific PC may act by directly affecting the function of ITR-1 or by changing membrane properties. Further studies are required to elucidate the underlying mechanism.

In summary, the present study uncovered a new cross-communication between stress response pathways and identified novel regulators of animal aging. Given the conservation of proteins and lipids identified, similar mechanisms of cellular homeostasis and longevity may be present in other species like mammals.

## Materials and methods

### *C. elegans* strains and maintenance

*C. elegans* were cultured on standard nematode growth medium (NGM) seeded with *E. coli* OP50-1 [49]. The following strains were provided by Caenorhabditis Genome Center: The

wild-type N2, AU78 [*T24B8.5p::gfp::unc-54*-3' UTR], CL2166[*gst-4p::gfp*], CF1553[*sod-3p::gfp*], TJ375[*hsp-16.2p::gfp*], TJ356 [*daf-16p::daf-16a/b::gfp*], CF1038[*daf-16(mu86)*], CB1370[*daf-2 (e1370)*], CB4037[*glp-1(e2141)*], JT73[*itr-1(sa73)*], VP303[*rde-1(ne219); kbIs7(nhx-2p::rde-1)*], WM118[*rde-1(ne300); neIs9(myo-3::ha::rde-1)*] and NR222[*rde-1(ne219); kzIs9(lin-26p::rde-1)*], MGH30[*sgk-1(mg455)*], MGH266[*rict-1(mg451)*], BQ1[*akt-1(mg306)*], AA18[*daf-12 (rh61rh412)*], RG1228[*daf-9(rh50)*], CF2167[*tcer-1(tm1452)*], F2052[*kri-1(ok1251)*]. The Strain DCL569 [*rde-1(mkc36); mkcSi13(sun-1p::rde-1::sun-1* 3'UTR)] was provided by Dr. Di Chen. The strain containing *dis-3::mcherry* was provided by Dr. Shouhong Guang. The strain SSP171[*mtl-1::gfp*] was generated by our own lab [50]. The strain SSP172[*let-607p::let-607::gfp*] was generated by cloning the 2.2kb promoter region and the full length of *let-607* genomic DNA into pPD95.79 plasmid that was injected into gonad by standard techniques.

## Microbe strains and RNA interference treatment

*E. coli* OP50-1 bacteria were cultured overnight at 37˚C in LB, after which 150ul of bacterial culture were seeded on 60mm NGM plates. For RNAi experiment, HT115 bacteria containing specific dsRNA-expression plasmids (Ahringer library) [51] were cultured overnight at 37˚C in LB containing 100ug/ml carbenicillin, and seeded onto NGM plates containing 5mM IPTG [52].

RNAi was induced at room temperature for 24 hours after seeding. Then, synchronized L1 worms were added to RNAi plates to knock down indicated genes. For *let-607* RNAi treatments, the bacterial cultures of target RNAi strains were diluted with the vector control at the ratios of 1:5.

## qRT-PCR

qRT-PCR was performed as previously described [53]. Briefly, L4 stage worms were collected, washed in M9 buffer and then homogenized in Trizol reagent (Life Technologies). RNAs were extracted according to the manufacturer's protocol. DNA contamination was digested with DNase I (Thermo Fisher Scientific) and RNA was subsequently reverse-transcribed to cDNA by using the RevertAid First Strand cDNA synthesis Kit (Thermo Fisher Scientific). Quantitative PCR was performed using SYBR Green (Bio-Rad). The expression of *snb-1* was used to normalize samples.

## Lifespan analysis

Lifespan assays were performed as previously described [12]. Briefly, synchronized L1 worms were added to NGM plates seeded with indicated RNAi strains. Lifespan analysis was performed at 20˚C, except for experiments involving *glp-1* mutants, which were kept at 25˚C from L1 to day 1 adult, and shifted to 20˚C thereafter. Worms were transferred every day during reproductive period. Worms that died of vulva burst, bagging or crawling off the plates were censored.

## Stress resistance assays

For TBHP (Sigma) resistance, day 1 adult worms were transferred to NGM plates supplemented with 10mM TBHP and incubated at 20˚C for survival analysis. For pathogen resistance, late L4 worms were transferred to plates containing *pseudomonas aeruginosa* strain PA14 and maintained at 25˚C for survival analysis. For heat shock resistance, day 1 adult worms were incubated at 35˚C for survival analysis. For DTT resistance, day 1 adult worms

were transferred to NGM plates supplemented with 8.5mM DTT and incubated at 20˚C for survival analysis.

## Fluorescent microscopy

To analyze GFP expression, worms were paralyzed with 2mM levamisole and fluorescent microscopic images were taken after mounted on slides. To study the DAF-16 nuclear localization, the levels of GFP nuclear localization were scored. Briefly, no nuclear GFP, GFP signal in the nucleus of anterior or posterior intestine cells and nuclear GFP in all intestinal cells are categorized as low, medium and high expression, respectively.

## RNA sequencing analysis

Total RNA from L4 stage worms was extracted using TRIzol reagent and used to generate sequencing libraries using the VAHTS Stranded mRNA-seq Library Prep Kit for Illumina. The quality and quantity of the libraries were determined using an Agilent Bioanalyzer 2100 (Agilent, USA) and ND-2000 (NanoDrop Technologies, USA), respectively. Only high-quality RNA samples (OD260/280 = 1.8~2.2, OD260/230$\geq$2.0, RIN$\geq$6.5, 28S:18S$\geq$1.0, >10μg) were used to construct sequencing library. The libraries were sequenced at a paired-end 150 bp read length on an Illumina HiseqX Ten. RNA-seq reads were aligned to the reference genome WBcel235 using Tophat v.2.0.6[54]. The differential gene and transcript expression analysis was performed using Cufflinks tools [55–57]. Genes with a fold change of > 2 (either up or down) and FDR of < 0.1 were considered as differentially regulated genes. Genes functional classification and GO term analysis were conducted by using Database for Annotation, Visualization, and Integrated Discovery (DAVID) version 6.8 [58]. For Genes functional classification, DAVID groups genes with similar annotation terms. Subsequently, the geometric mean (in -log scale) of EASE scores (p-value) of all terms in each group was calculated as enrichment score. The RNA sequencing data have been deposited in the GEO with an accession number of GSE155935.

## Fatty acid quantification

Fatty acid contents were measured as previously described [50,59]. 20,000 age-synchronized L4 worms were washed off plates and washed three times with water. Worm pellets were resuspended with 1.2 mL 2.5% $H_2SO_4$ in methanol and incubated at 80˚C for 1 hour. Then, 1ml supernatants were mixed with 1.2 ml hexane and 1.8 ml water to extract fatty acid methyl esters for GC-MS/MS analysis. The Supelco 37 Component FAME Mix (Sigma Aldrich) was used to determine the retention time. The *Shimadzu GCMS-TQ8040* Gas Chromatograph Mass Spectrometer equipped with a SH-Rxi-5sil MS column was used. Fatty acid contents were normalized to protein concentrations.

## Membrane lipid quantification

Thin-layer chromatography (TLC) was performed as previously described [59,60] with modifications. 50,000 L4 worms were collected and washed with M9 to remove bacteria, and were sonicated in 0.25 mL PBS. Then, 5 mL mixture of ice-cold chloroform:methanol (1:1) was added and mixed immediately. The solution was incubated overnight at -20˚C with occasional shaking to extract lipids. After incubation, 2.2 mL Hajra's solution (0.2M $H_3PO_4$; 1M KCl) was added. The lower lipid containing organic phase was recovered by centrifuging for 1 minute at 3,000 rpm. Lipids were dried under nitrogen and resuspended in chloroform for TLC separation.

The silica gel TLC plate was activated by incubating at 110˚C for 75 minutes. Samples were loaded onto the TLC plate alongside lipid standards. The plate was run with a chloroform: methanol:water:acetic acid solvent mixture (65:43:3:2.5) until the solvent front was three quarters of the way up the plate. The plate was subsequently dried and run with a new solvent mixture of hexane:diethyl ether:acetic acid (80:20:2) until the solvent front reached the top of the plate. The plates were sprayed with 0.005% primuline and visualized under UV light. Spots corresponding to the major phospholipids were scraped to tube and resuspended in 2.5% $H_2SO_4$ in methanol. Spots were incubated for 1 hour at 80˚C to create FAMEs for GC-MS/MS analysis. An internal standard of 50 μg of tridecanoic acid was added to each tube.

### Lipid supplementation

For treatment with ceramides, ceramides with various side chains were dissolved in ethanol to make 0.5 mg/mL stock solutions; 25 μg of each ceramide was added to the surface of NGM plates before bacterial seeding. For supplementation with PC, unsaturated PC were dissolved in DMSO to make 17 mM stock solutions; 51 nmol of each was added to the surface of NGM plates.

### Quantification and statistical analysis

Data were presented as mean ± SEM. Survival data were analyzed by using a log-rank (Mantel-Cox) test. The nuclear accumulation of DAF-16::GFP was analyzed by using a Chi-square and Fisher's exact test. Figs 1A, 2A, S3F and S6C were analyzed by using an unpaired student t-test. Figs 4B, S1A and S6B were analyzed by using a paired student t-test. Figs 5A, 5B, 5C, 5D and S6A were analyzed by using a multiple t-test followed by a Holm-Sidak post hoc test. Fig 2D was analyzed by using a One-way ANOVA. Other figures were analyzed by using Two-way ANOVA followed by a Tukey post hoc test. For Fig 2E, statistical significance of the overlap between two groups of genes was analyzed by using an online software provided by Jim Lund (http://nemates.org/MA/progs/overlap_stats.html). The probability (p) of finding an overlap of x genes are calculated, the statistical detail of which can be found at: http://nemates.org/MA/progs/representation.stats.html. $P < 0.05$ was considered as significant.

### Supporting information

**S1 Fig. LET-607 regulates stress responses.** (A) mRNA levels of *let-607* in *let-607* RNAi-treated worms (1:5 diluted) as measured by qPCR. n = 3 per group. (B) Comparison of LET-607 and CREBH protein sequences. (C) Effects of *let-607* RNAi on HSP-4::GFP expression induced by tunicamycin. Scale bar = 100 μm. (D-K) Effects of tissue-specific *let-607* RNAi on TBHP and heat stress resistance. (L-M) Effects of post-developmental *let-607* RNAi on TBHP resistance (L), heat resistance (M) and lifespan (N). Data were presented as mean ± SEM. *** $p < 0.001$.
(TIF)

**S2 Fig. Suppression of LET-607 activates stress response factor DAF-16.** (A-C) Effects of *let-607* RNAi on the expression of *hsp-16.2p*::GFP (A), *gst-4p*::GFP (B) and *sod-3p*::GFP (C). Scale bar = 100 μm. (D) Effects of ER UPR genes RNAi on their own expression and downstream targets. n = 3 per group. (E) Knockdown efficiencies of *aha-1* and *gsk-3* RNAi. n = 3 per group. (F-H) Effects of *aha-1* RNAi (F), *gsk-3* RNAi (G) and *nhr-49* mutation (H) on MTL-1::GFP expression. Upper panel, representative images. Lower panel: quantification data. Scale bar = 100 μm. Data were presented as mean ± SEM. * $p < 0.05$, *** $p < 0.001$.
(TIF)

**S3 Fig. Suppression of LET-607 increases heat stress resistance via HSF-1.** (A-B) Impact of *let-607* RNAi on pathogen resistance (A) and TBHP resistance (B) in WT and *hsf-1* mutant worms. (C-D) Effects of *let-607* RNAi on heat stress resistance in *daf-16* (C) and *hsf-1* (D) mutants. (E) Impact of *let-607* RNAi on the lifespan of *daf-2* mutants. (F) Effects of *let-607* RNAi on reproduction. Data were presented as mean ± SEM. *** $p < 0.001$.
(TIF)

**S4 Fig. SMS-5 mediates DAF-16 activation.** (A) Knockdown efficiency of *sms-5* RNAi. n = 3 per group. (B-C) Effects of *sms-5* RNAi on *let-607* RNAi-induced TBHP resistance (B) and longevity (C). (D) The schematic of SMS-5-mediated enzymatic reaction. (E) Supplementation of ceramides did not influence MTL-1::GFP expression with or without *let-607* RNAi. Left panel, representative images. Right panel: quantification data. Scale bar = 100 μm. Data were presented as mean ± SEM. *** $p < 0.001$.
(TIF)

**S5 Fig. PC mediates DAF-16 activation.** (A) The schematic of the PC biosynthetic pathway in *C. elegans*. (B) Knockdown efficiencies of PC biosynthetic genes RNAi. n = 3 per group. (C) Effects of PC biosynthetic genes RNAi on MTL-1::GFP expression in WT and *let-607* knock-down animals. Upper panel, representative images. Lower panel: quantification data. Scale bar = 100 μm. Data were presented as mean ± SEM. *** $p < 0.001$.
(TIF)

**S6 Fig. LET-607 regulates lipid metabolism.** (A) Effects of *let-607* RNAi on the abundance of total saturated and unsaturated fatty acids. (B) Total triglyceride levels after *let-607* RNAi treatment. (C) PC/PE ratio after *let-607* RNAi treatment. Data were presented as mean ± SEM. * $p < 0.05$, **$p < 0.01$, *** $p < 0.001$.
(TIF)

**S7 Fig. PC mediates DAF-16 nuclear occupancy in glp-1 mutants.** (A-B) Effects of PC (18:1n9) supplementation on DAF-16::GFP nuclear accumulation in *glp-1* mutants (A) and *daf-2* mutants (B). Scale bar = 100 μm. Upper panel shows representative images. White arrow indicates the nuclear GFP signal. Lower panel shows semi-quantification data. Number of animals (n) for (A): WT + control RNAi (85), WT+ *let-607* RNAi (85), *glp-1* + control RNAi (93), *glp-1* + *let-607* RNAi (109); number of animals (n) for (B): WT + control RNAi (65), WT+ *let-607* RNAi (65), *daf-2* + control RNAi (55) and *daf-2* + *let-607* RNAi (54). Scale bar = 50 μm. *** $p < 0.001$.
(TIF)

**S8 Fig. ITR-1-dependent calcium signaling mediates DAF-16 activation in response to let-607 knockdown.** (A-B) Knockdown efficiencies of *itr-1* RNAi (A) and *egl-8* RNAi (B). (C-D) Effects of *egl-8* RNAi on *let-607* RNAi-induced MTL-1::GFP expression (C) and TBHP resistance (D). Left panel, representative images. Right panel: quantification data. Scale bar = 100 μm. (E) Knockdown efficiencies of *sca -1* and *sms-5* RNAi in Fig 7D. (F) Effects of ionomycin on MTL-1::GFP expression suppressed by *sms-5* RNAi. Left panel, representative images. Right panel: quantification data. Scale bar = 100 μm. (G) Knockdown efficiencies of *pkc-2* RNAi. (H) Effects of *sgk-1* mutation on the TBHP resistance of WT and *let-607* knock-down worms. (I-J) Effects of *rict-1* (I) and *akt-1;akt-2 (RNAi)* (J) mutations on *let-607* RNAi-induced MTL-1::GFP expression. Left panel, representative images. Right panel: quantification data. For (A), (B), (E) and (G), n = 3 per group. Data were presented as mean ± SEM. *** $p < 0.001$.
(TIF)

**S1 Table. List of genes regulated by LET-607.**
(XLSX)

**S2 Table. PA14 survival data.**
(DOCX)

**S3 Table. TBHP survival data.**
(DOCX)

**S4 Table. 35˚C heat shock survival data.**
(DOCX)

**S5 Table. DTT survival data.**
(DOCX)

**S6 Table. Lifespan data.**
(DOCX)

**S7 Table. qPCR primer sequences.**
(DOCX)

## Acknowledgments

We thank CGC, Dr. Di Chen and Dr. Shouhong Guang for providing strains. We thank Qin Deng and Guocan Zheng in Analytic and Testing Center of Chongqing University for the use of facility and technical support.

## Author Contributions

**Conceptualization:** Shanshan Pang, Haiqing Tang.

**Data curation:** Bin He.

**Formal analysis:** Bin He, Jie Xu, Shanshan Pang, Haiqing Tang.

**Funding acquisition:** Shanshan Pang, Haiqing Tang.

**Investigation:** Bin He, Jie Xu.

**Methodology:** Bin He, Jie Xu.

**Project administration:** Shanshan Pang, Haiqing Tang.

**Resources:** Shanshan Pang, Haiqing Tang.

**Supervision:** Shanshan Pang, Haiqing Tang.

**Validation:** Bin He, Jie Xu.

**Writing – original draft:** Bin He, Shanshan Pang, Haiqing Tang.

**Writing – review & editing:** Shanshan Pang, Haiqing Tang.

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
