## [Decision Letter · Decision Letter 0]

13 Dec 2020

Dear Dr Tang,

Thank you very much for submitting your Research Article entitled 'Phosphatidylcholine mediates the crosstalk between ER and cytosolic stress response pathways' to PLOS Genetics.

The manuscript was fully evaluated at the editorial level and by independent peer reviewers. The reviewers appreciated the attention to an important problem, but raised some substantial concerns about the current manuscript. Based on the reviews, we will not be able to accept this version of the manuscript, but we would be willing to review a much-revised version. We cannot, of course, promise publication at that time.

If you decide to revise the manuscript for further consideration at PLOS Genetics, please aim to resubmit within the next 60 days, unless it will take extra time to address the concerns of the reviewers, in which case we would appreciate an expected resubmission date by email to plosgenetics@plos.org.

[LINK]

We are sorry that we cannot be more positive about your manuscript at this stage. Please do not hesitate to contact us if you have any concerns or questions.

Yours sincerely,

Coleen T. Murphy

Associate Editor

PLOS Genetics

Gregory P. Copenhaver

Editor-in-Chief

PLOS Genetics

Reviewer's Responses to Questions

**Comments to the Authors:**

Reviewer #1: In this study, He et al. identify that depletion of the let-607 transcription factor leads to activation of the stress-related transcription factor DAF-16, resulting in enhanced cytotoxic protection and lifespan extension. They term this induction of DAF-16 activity as the ER-to-cytosol stress pathways communication (ECSPC), based on the assumption that LET-607 is the C. elegans CREBH homolog, embedded in the ER membrane. They further show that let-607 depletion alters fatty acid metabolism, and specifically reduced the levels of unsaturated PC, which turned out to be important for DAF-16 activation under these conditions. The authors identified additional genes required for DAF-16 activation by let-607 depletion including the IP3 receptor itr-1, which releases Calcium stores from the ER to the cytoplasm and the calcium-dependent kinase PKC-2.

I find that the beneficial effects of let-607 inactivation on lifespan and stress resistance, the associated activation of DAF-16, and the alterations in lipid metabolism are all well established. Specifically, the regulation of DAF-16 by unsaturated PC is novel and of wide-interest. Based on these, I find this work interesting and novel.

That being said, the main take-home message conveyed by the authors is that this is a pathway that links between the cytosolic and the ER stress responses. Nevertheless, the connection to the ER is mainly based on the finding that its transcript levels rise upon ER stress and the annotation of let-607 as the C. elegans ortholog of CREBH (whose connection to the ER is well established). However, this statement is based on homology spanning only 19% of the protein, corresponding to the b-leucine zipper domain. I could not find any indication for the presence of a putative transmembrane domain in the LET-607 protein. Such a transmembrane domain would be essential for anchoring the protein to the ER. I understand that let-607 has been referred to as the C. elegans ortholog of CREBH in previous publications, but this still needs to be established, especially since this is the “major player” in this work. Thus, the authors should investigate the intercellular localization of LET-607. Is it indeed embedded in the ER or is it cytosolic/nuclear? If this is done by tagging the protein, be careful not to perturb any putative signal peptide. If this experiment does not demonstrate ER localization – the paper should be re-written accordingly. To clarify – this experiment is needed and should be included in the final manuscript whatever the result may be – to confirm the ER association of this protein or to prevent further confusion.

Additional major comments:

1) Epistatic analysis demonstrated that let-607 and germline depletion might act in the same or similar pathway as the lifespan effects were not additive. Likewise, both treatments lead to nuclear localization of DAF-16 specifically in the intestine. The question is – are these similar pathways that converge downstream, or does let-607 activate the germline pathway? This should be addressed in 2 ways: 1) What is the state of the germline in let-607 RNAi treated animals? 2) The longevity of glp-1 animals depends on unique genes such as daf-9, daf-12, kri-1, tcer-1 etc. Are these genes required for lifespan extension by let-607? Please do these experiments using mutants and let-607 RNAi (not double RNAis).

2) Does sms-5 RNAi/PC(18:1N9) supplementation interfere with other daf-16 activating pathways such as daf-2,glp-1, oxidative stress, heat shock?

3) Some of the experiments are done with multiple RNAis. This is especially problematic for epistasis analysis. These should be repeated using mutants when possible, or to the very least with a comparison of RNAi efficiencies.

Additional comments:

Page 8 – “LET-607, like its mammalian ortholog, could also be activated upon ER stress18” – activation has not been shown here, only increased transcript levels have been demonstrated.

Page 8 – “Gene functional classification analysis revealed that the LET-607-upregulated genes were enriched in stress responses” - I am not sure if stress responses are the best definition here. The explicit stress response GO categories did not show up in this analysis. Please word this more carefully.

Page 10 – “as RNAis of none of the ER UPR genes (ire-1, pek-1 and atf-6) had any effects on MTL-1::GFP expression” – negative result. Better done with mutants rather than RNAi.

Page 10 – “We compared the let-607 RNAi-induced genes with the DAF-16 class I targets, and found that 74 genes upregulated by let-607 RNAi (13.6%) belonged to the DAF-16 class I targets, which was significantly greater than that would be expected by chance7 (Figure S2E), further supporting that let-607 RNAi regulates the activity of DAF-16. ” Please comment whether these genes also contain LET-607 binding sites.

Figure 2 – number of animals scored is not the number of biological repeats.

A Summary model would be helpful

Reviewer #2: This manuscript by He et al. describes heretofore unappreciated genetic pathway connecting suppression of the ER stress factor CREBH/LET-607 to activation of a FoxO-dependent stress resistance pathway mediated by reduction in phosphatidylcholine containing fatty acids. The authors term this pathway the ER-to-cytosol stress pathways communication (ECSPC). The survival data are generally robustly gathered and I appreciate the inclusion of ~4 biological replicates for each of the conditions tested. Some of the mechanistic work based upon simultaneous knockdown of multiple genes at once with RNAi require more rigorous proof. Additionally, stronger genetic evidence for the involvement of LET-607 and factors previously reported to act in a pathway with LET-607 should be provided. I am generally positive on the novelty of the findings, but the work requires more rigor to substantiate a solid enough mechanism to garner publication in PLoS Genetics.

Although intelligible, the manuscript is written in broken English and absolutely requires proofreading by a native English speaker. There are too many errors to enumerate here.

There is no mention of significance for the calculated enrichment score and what the comparator group is for the analysis conducted in Figure 1B. Also, I assume that what the authors are referring to in Fig 1B when they say “LET-607-upregulated genes” is the set of genes upregulated when LET-607 is knocked down by RNAi. The manuscript would be well served if this was made more clear.

Are there other genes that activate the ECSPC in concert with CREBH, including previously reported partners/activators in CREBH/LET-607 activity BMAL1/AHA-1, GSK3b/GSK-3, or PPARalpha/NHR-49? Does genetic inactivation of the S1P/S2P also activate the ECSPC?

Does the nuclear form of CREBH/LET-607 act as a dominant suppressor of the ECSPC?

In addition to “n”, the tests used to calculate significance should be indicated in the main figure legends (for example but not limited to Figs. 2B, 2C), as well as significance for survival data (even if they are indicated with repeats in Table S2).

It is possible that the doses of PC(16:0) and PC(18:0) were insufficient to raise C. elegans levels of these metabolites and that is the reason they “fail” to suppress daf-16 responses in figure 3E. The authors should measure abundances of these species in treated animals along with the successful supplementation of PC(18:1n9) to substantiate the claim that it is unsaturated PC that is mediating the signaling pathway. Certainly because more than just 18:1n9 fatty acids are changing in the PC fraction of let-607 RNAi treated animals in figure 4D, even though, as the authors point out, the majority of changes are seen in PC unsaturated fatty acid abundances.

While I believe that the authors are likely correct on the conclusion of the involvement of calcium in the ECSPC activation, more evidence is needed to solidify some of the lines of evidence. Specifically, for the double RNAi experiments, the authors are already diluting let-607 RNAi to prevent lethality, how do they know, for example in Figure 5A/D/E/F that this is because they are no longer effectively knocking down let-607? This is not proven by the nicely done qPCR in Figure S1A. Further, I don’t know what to think when three RNAi are combined—this is generally not accepted as standard practice in the field without proof of equally good knockdown (i.e.figure 5D).

The demonstration of pkc-2 and sgk-1 dependency of the ECSPC are interesting and compelling, but the interpretation of these data are complicated. First, Sgk-1 mutants are viable, so consideration should be given to repeating the experiment in figure 5F with sgk-1 mutants, and not double RNAi. Second, while pkc-2 and sgk-1 are reported to be in a genetic pathway with each other with regard to cold survival in the worm, more canonical descending input into sgk-1 is through mTORC2. The authors should determine whether disruption of mTORC2 also recapitulates the sgk-1 result. Third, the best described input into daf-16 are the akt kinases. The authors should disrupt akt-1 and akt-2 by genetic mutation or RNAi and determine whether this mitigates activation of daf-16 in the ECSPC (e.g. in response to let-607 RNAi).

Minor comments

Please describe the pathogen stress used in the figure legend (Fig. 1C) or the text, as well as more detail about the stresses (i.e. duration, dose) used in the legend for figures 1C-1G.

Sphingomyelin on P.8 is misspelled.

**Have all data underlying the figures and results presented in the manuscript been provided?**

Reviewer #1: None

Reviewer #2: Yes

PLOS authors have the option to publish the peer review history of their article (what does this mean?). If published, this will include your full peer review and any attached files.

Reviewer #1: No

Reviewer #2: No

---

## [Decision Letter · Decision Letter 1]

16 Apr 2021

Dear Dr Tang,

Thank you very much for submitting your Research Article entitled 'Phosphatidylcholine mediates the crosstalk between LET-607 and DAF-16 stress response pathways' to PLOS Genetics.

The manuscript was fully evaluated at the editorial level and by independent peer reviewers. The reviewers appreciated the attention to an important topic but identified some concerns that we ask you address in a revised manuscript

We therefore ask you to modify the manuscript according to the review recommendations. Your revisions should address the specific points made by each reviewer.

[LINK]

Yours sincerely,

Coleen T. Murphy

Associate Editor

PLOS Genetics

Gregory P. Copenhaver

Editor-in-Chief

PLOS Genetics

Reviewer's Responses to Questions

**Comments to the Authors:**

Reviewer #1: I am overall pleased with the revised version of the manuscript, but still must insist on one follow-up experiment to clearly demonstrate the intra-cellular localization of LET-607.

I appreciate the attempts of the authors to determine let-607 localisation by generating a translational fusion. However, the provided images in Fig 1B are of poor quality, of low resolution and lack markers. From the little I can see, I think the localization may be ER after all, rather than nuclear. Better imaging is required. Higher resolution, confocal imaging, contrast labelling of the nuclei and an ER marker to examine possible colocalization

On the same note, DAF-16::gfp images are also of poor quality (Fig 2C. Fig 4C). Please provide better images.

Page 11, line 221 - since PC reduction alone did not activate DAF-16 - this sentence should be toned down.

Page 12, libe 250 - the suppression of mlt-1 expression in glp-1 mutants was only partial - tone down

Page 10, line 191 - Replace reproductivity with a better word

Page 14, line 271 - in defective in

Reviewer #2: This revised manuscript by He et al. for the most part addresses my concerns. I appreciate the addition of the data demonstrating interaction with the daf-2 and germlineless pathway, as well as the epistasis analyses with daf-9, daf-12, tcer-1 and kri-1. The PC-calcium pathway analyses are also compelling and well presented, though quantification of some of the reporter data would go a long way in strengthening the conclusions. It is an interesting story that is relevant to the broad readership of PLoS Genetics.

I have a few lingering concerns, mostly statistical with some minor points.

t-test is not appropriate for establishing significance when more than 1 test is being done, e.g. fig 2B where daf-16 and sod-3 expression are both tested—correction for multiple hypothesis testing is needed.

Paired student t-tests are not appropriate without multiple hypothesis testing for figures 5B, C, and F. B and C should be corrected as A and D for multiple hypothesis testing, and F should be compared by 2-way ANOVA with post-hoc tests for significance.

Minor points:

The authors use dilutional RNAi. Did they consider post-developmental RNAi to see if the lifespan extension and developmental lethality represent antagonistic pleiotropies?

Line 60: based upon their findings, the authors may want to say “In mammals, CREBH is an ER-bound transcription factor…”

Line 68 LET-607 “is located in the nucleus” or “localizes to the nucleus”

In figure S1B, membrane is spelled “membrance” in 2 places.

It may be useful, when making decisions about additivity of factors in stress pathway activation, to quantify data—e.g. fig S2f-H. Then again, is it really valuable to be making comments about additivity of hypomorphic effects by RNAi? A hypomorphic effect can be enhanced almost by definition. As such the analyses are not terribly informative as presented.

The authors may consider quantifying expression of mtl-1 to make conclusions about synthetic interactions in figs. 4 and S4.

Please state the statistical test used in Fig 2E.

The data in figure 6 are very interesting and while S6 is quantified, Fig 6 is not, especially in glp1 6A where a claim of dependence on PC is made. These data should be quantified.

Sgk-1 RNAi has been reported to be long lived (Baumeister lab, 2004 Dev Cell). Can the authors explain why their RNAi is not long lived in fig. 7G?

**Have all data underlying the figures and results presented in the manuscript been provided?**

Reviewer #1: Yes

Reviewer #2: Yes

PLOS authors have the option to publish the peer review history of their article (what does this mean?). If published, this will include your full peer review and any attached files.

Reviewer #1: No

Reviewer #2: No

---

## [Editor Report · Decision Letter 2]

3 May 2021

Dear Dr Tang,

We are pleased to inform you that your manuscript entitled "Phosphatidylcholine mediates the crosstalk between LET-607 and DAF-16 stress response pathways" has been editorially accepted for publication in PLOS Genetics. Congratulations!

Yours sincerely,

Coleen T. Murphy

Associate Editor

PLOS Genetics

Gregory P. Copenhaver

Editor-in-Chief

PLOS Genetics

Comments from the reviewers (if applicable):

The authors have addressed the reviewers' remaining concerns.

**Data Deposition**

http://datadryad.org/submit?journalID=pgenetics&manu=PGENETICS-D-20-01382R2

**Press Queries**

---

## [Editor Report · Acceptance letter]

17 May 2021

PGENETICS-D-20-01382R2 

Phosphatidylcholine mediates the crosstalk between LET-607 and DAF-16 stress response pathways 

Dear Dr Tang, 

We are pleased to inform you that your manuscript entitled "Phosphatidylcholine mediates the crosstalk between LET-607 and DAF-16 stress response pathways" has been formally accepted for publication in PLOS Genetics! Your manuscript is now with our production department and you will be notified of the publication date in due course.

With kind regards,

Katalin Szabo

PLOS Genetics

On behalf of:
